# FedOBP: Federated Optimal Brain Personalization with Few Personalized Parameters

## Abstract

Personalized Federated Learning (PFL) addresses the challenge of data heterogeneity across clients by adapting global knowledge to local data distributions. A promising approach within PFL is model decoupling, which separates the Federated Learning (FL) model into global and personalized parameters. Consequently, a key question in PFL is determining which parameters should be personalized to balance global knowledge sharing and local data adaptation. In this paper, we propose a parameter decoupling algorithm with a quantile-based thresholding mechanism and introduce an element-wise importance score, termed Federated Optimal Brain Personalization (`FedOBP`). This score extends Optimal Brain Damage (OBD) pruning theory by incorporating a federated approximation of the first-order derivative in the Taylor expansion to evaluate the importance of each parameter for personalization. Extensive experiments demonstrate that `FedOBP` outperforms state-of-the-art methods across diverse datasets and heterogeneity scenarios, while requiring personalization of only a very small number of personalized parameters.

## 1 Introduction

Federated Learning (McMahan et al., 2017) is a distributed paradigm that facilitates collaborative model training across multiple clients while preserving the decentralized nature of their data. However, data heterogeneity among clients, often characterized by non-independent and identically distributed (non-IID), typically leads to sub-optimal performance (Karimireddy et al., 2020).

To address this challenge, personalized Federated Learning (PFL) enables individual clients to maintain customized models tailored to their local data distributions while also sharing knowledge across clients (Collins et al., 2022). A representative type of PFL algorithm is model decoupling (Collins et al., 2021; OH et al., 2022; Xu et al., 2023; Xingyan et al., 2024; Mclaughlin & Su, 2024), which divides the neural network model into globally shared feature extractor and a locally personalized prediction head. An extension of this approach, parameter decoupling (Yang et al., 2023; Zhou et al., 2024; Tamirisa et al., 2024), provides a finer-grained decomposition by dividing model parameters into element-wise global and personalized subsets.

A critical challenge of parameter decoupling is identifying which parameters should be personalized. Recent studies have proposed various approaches to address this issue. For example, FedSelectTamirisa et al. (2024) and PSPFLZhou et al. (2024) suggest that personalized parameters should be characterized by higher parameter changes (e.g., accumulated gradients) during local training. Meanwhile, FedDPA (Yang et al., 2023) identifies parameters with larger Fisher information values as more suitable for personalization. However, these methods often lack a solid theoretical foundation for selecting personalized parameters.

In this paper, we propose a novel PFL framework and an element-wise importance score, named Federated Optimal Brain Personalization (`FedOBP`) that achieves strong performance with very few personalized parameters. The main contributions of this work are as follows:

- We introduce a parameter decoupling algorithm with a quantile-based thresholding mechanism, selecting a small subset of personalized parameters to replace the global parameters for each client.

- We propose a scoring function `FedOBP` for personalized parameter selection. This score extends Optimal Brain Damage (OBD) pruning theory (LeCun et al., 1989) by incorporating a federated approximation of the first-order derivative within the Taylor expansion.

- We perform extensive experiments demonstrating that our method achieves state-of-the-art performance with only a small number of personalized parameters ($< 0.5\%$), ensuring both efficiency and effectiveness in personalization.

## 2 RELATED WORKS

**Personalized Federated Learning (PFL)**   PFL has been studied from multiple perspectives, focusing on addressing data heterogeneity and enhancing model performance. For instance, APFL (Deng et al., 2020) allows clients to train local models while contributing to the global model by adaptively combining local and global parameters. FedPer (Arivazhagan et al., 2019) introduces a layer-wise decoupling design, separating base and personalized layers to address data heterogeneity. LG-FedAvg (Liang et al., 2020) takes the opposite approach by training the feature extractor locally and the classifier globally to mitigate the effects of data heterogeneity. In contrast, FedRep (Collins et al., 2021) trains the feature extractor globally while training the classifier locally to tackle heterogeneity issues. pFedFDA (Mclaughlin & Su, 2024) addresses the limitations of transitional layer-wise model decoupling, particularly the bias-variance trade-off in classifier training, which relies solely on local datasets. It also views classifier representation learning as a generative modeling task, training representations based on the global feature distribution. FLOCO (Grinwald et al., 2024) leverage linear mode connectivity to identify a linearly connected low-loss region within the parameter space of neural networks. This approach allows clients to personalize their local models within designated subregions, while simultaneously collaborating to train a global model. FLUTE (Liu et al., 2024) consider federated representation learning under-parameterized regime, which integrates low-rank matrix approximation techniques with FL analysis. Some other methods (Chen & Chao, 2022; Tan et al., 2023) also treat the model as a global feature extractor and a personalized classifier head.

**Personalized Parameter Importance Score**   Building on the concept of model decoupling, some approaches use element-wise scoring strategies to identify personalized parameters. For instance, FedSelect (Tamirisa et al., 2024) and PSPFL (Zhou et al., 2024) identify the model parameters with large local training updates for personalization. In contrast, FedDPA (Yang et al., 2023) employs Fisher information-based scoring to assess the sensitivity of each parameter, thereby identifying which parameters should be personalized.

**Optimal Brain Damage (OBD)**   Optimal Brain Damage (LeCun et al., 1989) and Optimal Brain Surgery (OBS) (Hassibi & Stork, 1992) quantify parameter importance primarily through a second-order Taylor expansion of the loss function. OBD has been well-established in theory and validated across various fields, including large language models (Ma et al., 2023; Zhang et al., 2023b). However, to the best of our knowledge, its application to PFL remains unexplored due to the complexities of the distributed learning paradigm. In this work, one of our key contributions is extending OBD by incorporating a federated approximation for personalized parameter selection in PFL.

## 3 METHOD

### 3.1 PRELIMINARY

**Federated Learning (FL)**   We consider the FL training process to consist of $T$ communication rounds. In each round $t$, a subset of clients $\mathcal{C}^t \subset \mathcal{C}$ is selected, where $|\mathcal{C}^t| = \gamma \cdot |\mathcal{C}|$ denotes the number of participating clients, with $t \in [1, T]$ and the participation rate $\gamma$. Initially, the server distributes the initialized model $\boldsymbol{\theta}_g^0$ to selected clients. These clients then perform local training using their client-specific datasets $\mathcal{D}_i$. The local objective for each client $i \in \mathcal{C}^t$ is to minimize the empirical loss over its local dataset $\mathcal{D}_i$:

$$\arg\min_{\boldsymbol{\theta}_i} \mathcal{L}(\boldsymbol{\theta}_i; \mathcal{D}_i) = \mathbb{E}_{(x,y)\sim\mathcal{D}_i}[\ell(\boldsymbol{\theta}_i; (x, y))], \tag{1}$$

where $\ell(\boldsymbol{\theta}_i; (x, y))$ is the loss function for a sample $(x, y)$ with parameters $\boldsymbol{\theta}_i$, and $\mathcal{L}(\boldsymbol{\theta}_i; \mathcal{D}_i)$ is the expected loss on dataset $\mathcal{D}_i$. After local training, clients send their updated models $\{\boldsymbol{\theta}_i^t\}_{i\in\mathcal{C}^t}$ back to

the server for the global aggregation:

$$\boldsymbol{\theta}_g^t = \sum_{i \in \mathcal{C}^t} \frac{m_i^t}{m^t} \boldsymbol{\theta}_i^t, \tag{2}$$

where $m^t = \sum_{i \in \mathcal{C}^t} m_i^t$ and $m_i^t$ is the local sample count of client $i$ at $t$. The aggregated parameters $\boldsymbol{\theta}_g^t$ are then distributed to the local clients for the next communication round. Therefore, the global objective can be expressed as:

$$\arg\min_{\boldsymbol{\theta}} \mathcal{L}(\boldsymbol{\theta}_g; \mathcal{D}) = \frac{1}{|\mathcal{C}|} \sum_{i \in \mathcal{C}} \mathbb{E}_{(x,y) \sim \mathcal{D}_i}[\ell(\boldsymbol{\theta}_g; (x, y))], \tag{3}$$
$$= \mathbb{E}_{(x,y) \sim \mathcal{D}}[\ell(\boldsymbol{\theta}_g; (x, y))],$$

where $\mathcal{L}(\boldsymbol{\theta}_g; \mathcal{D})$ represents the expected loss over the entire dataset $\mathcal{D} = \{\mathcal{D}_i\}_{i \in \mathcal{C}}$ across all clients.

**Model Decoupling**    Model Decoupling addresses data distribution heterogeneity by selecting a client-specific personalized subset $\boldsymbol{u}_i^t$ from the previous local model $\boldsymbol{\theta}_i^{t-1}$ and choosing a globally shared subset $\boldsymbol{v}_i^t$ from the global model parameters $\boldsymbol{\theta}_g^t$. Client $i$ combines the client-specific personalized subset $\boldsymbol{u}_i^t$ with the globally shared subset $\boldsymbol{v}_i^t$ to create the merged model $\tilde{\boldsymbol{\theta}}_i^t = \{\boldsymbol{u}_i^t, \boldsymbol{v}_i^t\}$. Let $\mathcal{K}$ denote the set of all parameter indices, such that $\boldsymbol{\theta}_g^t = \{\theta_g^{t,k}\}_{k \in \mathcal{K}}$. For subsets $\boldsymbol{u}_i^t$ and $\boldsymbol{v}_i^t$, their corresponding parameter index sets are $\mathcal{K}(\boldsymbol{u}_i^t)$ and $\mathcal{K}(\boldsymbol{v}_i^t)$ respectively, where $\boldsymbol{u}_i^t = \{\theta_i^{t-1,k}\}_{k \in \mathcal{K}(\boldsymbol{u}_i^t)}$ and $\boldsymbol{v}_i^t = \{\theta_g^{t,k}\}_{k \in \mathcal{K}(\boldsymbol{v}_i^t)}$. It is important to note that the element-wise parameter decoupling for each parameter $k$ can vary across clients $i$ and communication rounds $t$. While $\boldsymbol{u}_i^t$ is updated exclusively using the client's local dataset $\mathcal{D}_i$, $\boldsymbol{v}_i^t$ is involved in both local updates and global parameter aggregation. The local objective function for PFL with parameter decoupling can be expressed as:

$$\arg\min_{\{\tilde{\boldsymbol{\theta}}_i^t\}_{i \in \mathcal{C}}} \left\{ \frac{1}{|\mathcal{C}|} \sum_{i \in \mathcal{C}} \mathcal{L}(\tilde{\boldsymbol{\theta}}_i^t; \mathcal{D}_i) \right\}, \tag{4}$$

where $\mathcal{L}(\tilde{\boldsymbol{\theta}}_i^t; \mathcal{D}_i) = \mathbb{E}_{(x,y) \sim \mathcal{D}_i}[\ell(\tilde{\boldsymbol{\theta}}_i^t; (x, y))]$ represents the expected loss for client $i$ with its decoupled parameters. $\ell(\tilde{\boldsymbol{\theta}}_i^t; (x, y))$ is the loss function for a sample $(x, y)$ computed using the client-specific model $\tilde{\boldsymbol{\theta}}_i^t$.

**Parameter Importance Score**    PFL with parameter decoupling methods (Yang et al., 2023; Zhou et al., 2024; Tamirisa et al., 2024) identify the personalized parameter set $\boldsymbol{u}_i^t$ and the globally shared parameter set $\boldsymbol{v}_i^t$ based on an element-wise parameter importance score $I(\cdot)$.

FedSelect (Tamirisa et al., 2024) and PSPFL (Zhou et al., 2024) propose using local updates from the pre-trained merged model $\tilde{\boldsymbol{\theta}}_i^{t-1}$ in the previous round to compute the importance score of each parameter $\theta_i^{t-1,k}, k \in \mathcal{K}$. This approach relies on multiple local updates and gradient computations during local training. Specifically, the gradient-based importance score $I_G(\cdot)$ is calculated as the absolute difference between the merged model $\tilde{\theta}_i^{t-1,k}$ and the locally updated $\theta_i^{t-1,k}$ as follows:

$$I_{\mathrm{G}}\left(\theta_i^{t-1,k}; \mathcal{D}_i\right) = \left|\tilde{\theta}_i^{t-1,k} - \theta_i^{t-1,k}\right|. \tag{5}$$

However, gradient-based importance scores can only be determined after local training in the previous communication round $t-1$. This limitation means that the score can only be used if the client is selected again in subsequent rounds, introducing a delay issue when the participation rate $\gamma < 1$.

FedDPA (Yang et al., 2023) takes a different approach by utilizing Fisher information to determine importance for personalization. The Fisher information-based importance score $I_F(\cdot)$ is defined as:

$$I_{\mathrm{F}}\left(\theta_i^{t-1,k}; \mathcal{D}_i\right) = \left(\frac{\partial \mathcal{L}(\theta_i^{t-1,k}, \mathcal{D}_i)}{\partial \theta_i^{t-1,k}}\right)^2. \tag{6}$$

Unlike gradient-based methods, the Fisher information-based approach can be applied before local training and does not suffer from the delay issue. However, it still requires gradient computation in order to calculate the Fisher information for each parameter $\{\theta^{t-1,k}\}_{k \in \mathcal{K}}$ with the local dataset $\mathcal{D}_i$,

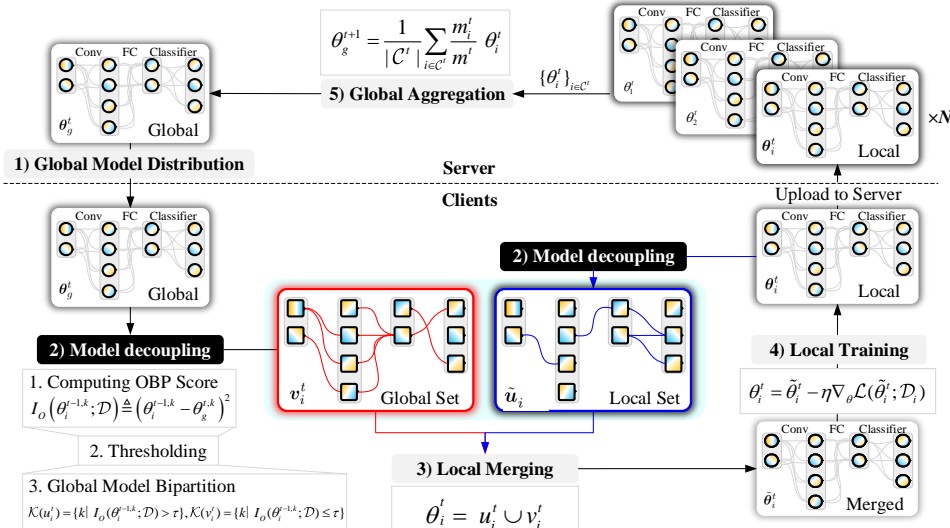

Figure 1: Overview of FedOBP. The server sends a global model $\boldsymbol{\theta}_g^t$ to selected clients $\mathcal{C}^t$. Each client $i$ determines the personalized parameters based on the Federated OBP parameter importance $I_O(\boldsymbol{\theta}_i^{t-1}; \mathcal{D})$ using a quantile-based thresholding. The client $i$ merges the globally shared parameters $\boldsymbol{v}_i^t$ and the personalized parameters $\boldsymbol{u}_i^t$ to form the merged model $\tilde{\boldsymbol{\theta}}_i^t$. The client $i$ performs local training on the merged model $\tilde{\boldsymbol{\theta}}_i^t$ and uploads the updated model $\boldsymbol{\theta}_i^t$ to the server for global aggregation.

which introduces additional computational overhead. Furthermore, we observe that both the gradient-based and Fisher information-based methods require a relatively large proportion of personalized parameters to achieve optimal performance. This requirement for a large proportion of personalized parameters limits the ability of local models to effectively leverage shared knowledge across clients, thereby reducing the overall benefits of collaboration.

## 3.2 FEDOBP ALGORITHM

**Algorithm** The workflow of the FedOBP algorithm is illustrated in Figure 1. FedOBP follows the general framework of standard PFL, incorporating Federated OBP parameter importance score function $I_O(\cdot)$. The pseudocode of the main steps is provided in Algorithm 1.

*1) Global Model Distribution:* The server transmits the globally aggregated model $\boldsymbol{\theta}_g^t$ to the selected clients $i \in \mathcal{C}^t$.

*2) Model Decoupling:* Each client $i$ adopts a thresholding mechanism based on the Federated OBP parameter importance $I_O(\boldsymbol{\theta}_i^{t-1}; \mathcal{D}) = \{I_O(\boldsymbol{\theta}_i^{t-1,k}; \mathcal{D})\}_{k \in \mathcal{K}}$ to decouple the personalized set $\boldsymbol{u}_i^t$ from the previous local model $\boldsymbol{\theta}_i^{t-1}$ and the globally shared subset $\boldsymbol{v}_i^t$ from the global model $\boldsymbol{\theta}_g^t$. We will provide the detail of Federated OBP parameter importance function in Section 3.3. The importance-based partitioning uses a quantile-based thresholding function $f_q : I_O(\boldsymbol{\theta}_i^{t-1,k}; \mathcal{D}) \to \mathbb{R}$, which determines a threshold $\tau$, as follows:

$$\tau = f_q(I_O(\boldsymbol{\theta}_i^{t-1}; \mathcal{D}))$$
$$\triangleq \inf\{I_O(\boldsymbol{\theta}_i^{t-1}; \mathcal{D}) \in \mathbb{R} : F(I_O(\boldsymbol{\theta}_i^{t-1}; \mathcal{D})) \geq q\},$$

where $F(x)$ is the cumulative distribution function of $I_O(\boldsymbol{\theta}_i^{t-1}; \mathcal{D})$ and $q \in [0, 1]$ is the quantile level.

Using this threshold $\tau$, we can determine the personalized parameter set $\boldsymbol{u}_i^t = \{\theta_i^{t-1,k}\}_{k \in \mathcal{K}(\boldsymbol{u}_i^t)}$ and the globally shared parameter set $\boldsymbol{v}_i^t = \{\theta_g^{t,k}\}_{k \in \mathcal{K}(\boldsymbol{v}_i^t)}$ with the corresponding index sets as follows:

$$\mathcal{K}(\boldsymbol{u}_i^t) = \{k \mid I_O(\boldsymbol{\theta}_i^{t-1,k}; \mathcal{D}) > \tau\}, \tag{7}$$
$$\mathcal{K}(\boldsymbol{v}_i^t) = \{k \mid I_O(\boldsymbol{\theta}_i^{t-1,k}; \mathcal{D}) \leq \tau\}, \tag{8}$$

where $\theta_i^{t-1,k}$ represents the previous local model parameter at position $k$, and $\tau$ is a threshold.

---

**Algorithm 1: `FedOBP` $(T, \gamma)$**

---

**Input:** Total rounds $T$, participation rate $\gamma$.
**Output:** Global model $\boldsymbol{\theta}_g^T$, local models $\{\boldsymbol{\theta}_i^T\}_{i \in \mathcal{C}}$

1   Initialize $\boldsymbol{\theta}_g^0$;
2   **for** $t = 1$ **to** $T$ **do**
3      Distribute $\boldsymbol{\theta}_g^t$ to selected clients $\mathcal{C}^t$, $|\mathcal{C}^t| = \gamma|\mathcal{C}|$;
4      **foreach** $i \in \mathcal{C}^t$ *in parallel* **do**
5          Compute `FedOBP` parameter importance $\{I_O(\theta_i^{t-1,k}; \mathcal{D})\}_{k \in \mathcal{K}}$ based on Eq. equation 19;
6          Decouple $\{\boldsymbol{u}_i^t, \boldsymbol{v}_i^t\}$ based on Eq. equation 7 equation 8;
7          Merge model $\tilde{\boldsymbol{\theta}}_i^t = \boldsymbol{u}_i^t \cup \boldsymbol{v}_i^t$;
8          Training $\boldsymbol{\theta}_i^t \leftarrow \tilde{\boldsymbol{\theta}}_i^t - \eta\nabla_{\boldsymbol{\theta}}\mathcal{L}(\tilde{\boldsymbol{\theta}}_i^t; \mathcal{D}_i)$;
9      **end**
10      Aggregate $\{\boldsymbol{\theta}_i^t\}_{i \in \mathcal{C}^t}$ based on Eq. equation 11 to get $\boldsymbol{\theta}_g^{t+1}$;
11      Send $\boldsymbol{\theta}_g^{t+1}$ to the next round client $i \in \mathcal{C}^{t+1}$;
12   **end**
13   **return** $\boldsymbol{\theta}_g^T, \{\tilde{\boldsymbol{\theta}}_i^T\}_{i \in \mathcal{C}}$;

---

Through extensive numerical experiments in Section 4.2, we found that `FedOBP` can achieve strong performance by selecting very few personalized parameters $\boldsymbol{u}_i$. Notably, most selected personalized parameters are concentrated in the classifier layer, aligning with theoretical insights from Centered Kernel Alignment (CKA) (Hinton et al., 2015).

*3) Local Merging:* Each client $i$ obtains the merged model $\tilde{\boldsymbol{\theta}}_i^t$ by combining the client-specific personalized parameter subset $\boldsymbol{u}_i^t$ with the global parameter subset $\boldsymbol{v}_i^t$ as follows:

$$\tilde{\boldsymbol{\theta}}_i^t = \boldsymbol{u}_i^t \cup \boldsymbol{v}_i^t. \tag{9}$$

*4) Local Training:* Each client $i$ performs local training on the merged model $\tilde{\boldsymbol{\theta}}_i^t$ with the local dataset $\mathcal{D}_i$ to obtain the locally trained model $\boldsymbol{\theta}_i^t$, following the update rule:

$$\boldsymbol{\theta}_i^t = \tilde{\boldsymbol{\theta}}_i^t - \eta\nabla_{\boldsymbol{\theta}}\mathcal{L}(\tilde{\boldsymbol{\theta}}_i^t; \mathcal{D}_i), \tag{10}$$

where $\eta$ is the learning rate. In FL, such model updates can be performed multiple times. Then, each client $i$ uploads its locally trained model $\boldsymbol{\theta}_i^t$ to the server.

*5) Global Aggregation:* The server aggregates the locally trained models $\{\boldsymbol{\theta}_i^t\}_{i \in \mathcal{C}^t}$ to compuate the globally aggregated model $\boldsymbol{\theta}_g^{t+1}$ for the next communication round:

$$\boldsymbol{\theta}_g^{t+1} = \frac{1}{|\mathcal{C}^t|}\sum_{i \in \mathcal{C}^t}\frac{m_i^t}{m^t}\boldsymbol{\theta}_i^t. \tag{11}$$

### 3.3   FEDERATED OBP PARAMETER IMPORTANCE

**Optimal Brain Damage (OBD)**   Optimal Brain Damage (LeCun et al., 1989; Hassibi & Stork, 1992; Molchanov et al., 2019; Zhang et al., 2023b; Ma et al., 2023) is a model pruning technique that quantifies the element-wise importance of each parameter $\theta^k$ in a model $\boldsymbol{\theta} = \{\theta^k\}_{k \in \mathcal{K}}$ with respect to the loss function $\mathcal{L}(\cdot)$ on dataset $\mathcal{D}$ as:

$$I_O(\theta^k; \mathcal{D}) = |\Delta\mathcal{L}(\theta^k; \mathcal{D})| = |\mathcal{L}(\theta_{=0}^k; \mathcal{D}) - \mathcal{L}(\theta^k; \mathcal{D})|, \tag{12}$$

where $\mathcal{L}(\theta_{=0}^k; \mathcal{D})$ represents the expected loss with parameter $\theta^k$ is set to 0. The importance $I_O(\theta^k; \mathcal{D})$ of parameter $\theta^k$ at position $k$ can be further expanded using a Taylor series approximation of $\mathcal{L}(\theta_{=0}^k; \mathcal{D})$ at $\theta^k$ to obtain:

$$I_O(\theta^k; \mathcal{D}) = \left| \frac{\partial\mathcal{L}(\theta^k; \mathcal{D})}{\partial\theta^k}\delta\theta^k + \frac{1}{2}\delta\theta^k H_{kk}\delta\theta^k + \mathcal{O}(\|\theta^k\|^3) \right|, \delta\theta^k = \theta_{=0}^k - \theta^k = -\theta^k \tag{13}$$

where $H_{kk}$ is the diagonal entry of the Hessian matrix, capturing the second-order curvature of $\mathcal{L}(\cdot)$ with respect to $\theta^k$. $\mathcal{O}(\|\theta^k\|^3)$ denotes higher-order terms. Classical OBD and Optimal Brain Surgeon (OBS) methods primarily focus on pruning models after convergence relying on the second order term of the Taylor series approximation while assuming the first order term to be negligible (LeCun et al., 1989; Hassibi & Stork, 1992).

**Federated Optimal Brain Personalization (`FedOBP`)** As discussed in Section 3.1, global aggregation optimizes the global model $\boldsymbol{\theta}_g^t$ by minimizing the global loss equation 3 on the global dataset $\mathcal{D}$. However, due to data heterogeneity across clients, this aggregation can degrade the performance of each personalized model $\boldsymbol{\theta}_i^t$ on its corresponding local dataset $\mathcal{D}_i$. To maximize the benefits of shared global knowledge, we aim to share most parameters across clients and personalize very few parameters. To achieve this, we choose to identify the local parameters $\theta_i^{t-1,k}$ that are most critical for the global dataset $\mathcal{D}$.

Building on the OBD pruning theory, we introduce the Federated Optimal Brain Personalization (`FedOBP`) score function to assess the importance of each local model parameter $\theta_i^{t-1,k}$ with respect to the corresponding global model parameter $\theta_g^{t,k}$ for the global dataset $\mathcal{D}$. We use a Taylor series approximation of $\mathcal{L}(\theta_g^{t,k}; \mathcal{D})$ at $\theta_i^{t-1,k}$ to obtain:

$$I_O\left(\theta_i^{t-1,k}; \mathcal{D}\right) = \left| \mathcal{L}\left(\theta_g^{t,k}; \mathcal{D}\right) - \mathcal{L}\left(\theta_i^{t-1,k}; \mathcal{D}\right) \right| \tag{14}$$

$$= \left| \frac{\partial \mathcal{L}(\theta_i^{t-1,k}; \mathcal{D})}{\partial \theta_i^{t-1,k}} \delta\theta_i^{t-1,k} + \frac{1}{2}\delta\theta_i^{t-1,k} H_{kk} \delta\theta_i^{t-1,k} + \mathcal{O}(\|\theta^{t-1,k}\|^3) \right|, \tag{15}$$

where $\delta\theta_i^{t-1,k} = \theta_g^{t,k} - \theta_i^{t-1,k}$. In classical OBD, the importance of parameters is measured by the loss difference between pruned and unpruned parameters. In the proposed `FedOBP`, the importance of global parameters with respect to the global dataset is measured by the loss difference between local and global parameters.

`FedOBP` applies parameter importance analysis during the FL training phase, where the first order term often dominates the second order term in magnitude. As a result, `FedOBP` approximates parameter importance using only the first order term (Molchanov et al., 2019; Zhang et al., 2023b):

$$I_O\left(\theta_i^{t-1,k}; \mathcal{D}\right) \approx \left| \frac{\partial \mathcal{L}(\theta_i^{t-1,k}; \mathcal{D})}{\partial \theta_i^{i-1,k}} \cdot (\theta_g^{t,k} - \theta_i^{t-1,k}) \right| \tag{16}$$

Meanwhile, we interpret the global aggregation process in Eq. equation 11 as performing a single update step for the local models $\{\boldsymbol{\theta}_i^{t-1}\}_{i \in \mathcal{C}^t}$ on the global dataset $\mathcal{D}$. Consequently, the gradient descent formulation of the global aggregation for each parameter $\theta_i^{t-1,k}$ can be expressed as:

$$\theta_g^{t,k} \approx \theta_i^{t-1,k} - \eta \frac{\partial \mathcal{L}(\theta_i^{t-1,k}; \mathcal{D})}{\partial \theta_i^{i-1,k}}. \tag{17}$$

According to the classical FedAvg, multi-step cumulative gradient updates provide more accurate parameter improvements than a single-step gradient descent update (McMahan et al., 2017). A well-known federated optimization approach, Adaptive Federated Optimization (AFO) (Reddi et al., 2021), also interprets global aggregation as one step of gradient descent. Therefore, the federated gradient can be regarded as global-local parameter update as follows:

$$\frac{\partial \mathcal{L}(\theta_i^{t-1,k}; \mathcal{D})}{\partial \theta_i^{t-1,k}} \approx \theta_i^{t-1,k} - \theta_g^{t,k}. \tag{18}$$

Unlike AFO, which treats sequential global aggregations $(\theta_g^{t-1,k} - \theta_g^{t,k})$ as "gradient", we regard the federated global-local parameter update $(\theta_i^{t-1,k} - \theta_g^{t,k})$ in PFL as an approximation to the gradient. Therefore, we can further approximate Eq. equation 16 by replacing the gradient term with the parameter update $(\theta_i^{t-1,k} - \theta_g^{t,k})$ in Eq. equation 18 and obtain the `FedOBP` score function as follows:

$$I_O\left(\theta_i^{t-1,k}; \mathcal{D}\right) \triangleq \left(\theta_i^{t-1,k} - \theta_g^{t,k}\right)^2. \tag{19}$$

`FedOBP` introduces a interpretable and practical criterion by directly quantifying the discrepancy between local and global parameters for the global dataset. This formulation enables a more principled selection of parameters for personalization, allowing clients to identify and retain only the most impactful parameters based on their relevance to global knowledge for efficient personalization. Specifically, local parameters $\theta_i^{t-1,k}$ with higher `FedOBP` importance scores $I_O(\theta_i^{t-1,k}; \mathcal{D})$ are the most influential in improving global model performance. Personalizing these parameters is crucial to avoid excessive alignment with the global model, thereby preserving strong local performance. This allows clients to personalize a minimal and critical subset of parameters while sharing the rest globally, achieving an effective balance between local adaptation and collaborative learning. Extensive experiments show that this OBD-based personalization strategy consistently achieves better trade-offs between generalization and personalization, outperforming gradient- and Fisher-based methods across diverse benchmarks.

## 4 EXPERIMENTS

### 4.1 EXPERIMENTAL SETUP

**Datasets** We evaluate the propose method on four benchmark datasets, where EMNIST (Cohen et al., 2017) covers 62-class handwriting image classification, CIFAR10 and CIFAR100 (Krizhevsky, 2009) are with 10 and 100 classes respectively. SVHN (Netzer et al., 2011) focuses on 10-class digit classification. Data was evenly split into non-overlapping train and test sets per client, with heterogeneity simulated using $\text{Dir}(\alpha)$, $\alpha \in \{0.1, 0.5\}$ distribution (lower values indicating higher heterogeneity). The federated setup has global communication rounds $T = 400$ across 100 clients with participation rate $\gamma = 0.1$. Clients used Stochastic Gradient Descent optimization with a learning rate $\eta = 0.01$, 32 batch size, and 5 local epochs. All methods were evaluated over four random experiments, and the mean and standard deviation of the results were reported. Additional experimental details and results are detailed in the Appendix B.

**Baselines** We implement the baselines based on an open-source benchmarkTan & Wang (2025). We compare the performance of our `FedOBP` algorithm with eight current PFL methods, including FedPer (Arivazhagan et al., 2019), APFL (Deng et al., 2020), LG-FedAvg (Liang et al., 2020), FedRep (Collins et al., 2021), pFedFDA (Mclaughlin & Su, 2024), FLUTE (Liu et al., 2024), FedDPA (Yang et al., 2023), FLOCO (Grinwald et al., 2024) and FedALA (Zhang et al., 2023a). FedAvg (McMahan et al., 2017) and Local-Only served as baselines for assessing generalization and personalization performance.

**Model and Hyperparameters** We use a simple 4-layer CNN model with two convolutional layers and two fully connected (FC) layers, with the final FC layer serving as the classifier. The details of the CNN model is shown in Appendix B. In addition, we introduce two normalization strategies for parameter importance (LayerNorm and GlobalNorm). Specifically, LayerNorm applies a layer-wise min-max normalization procedure as (Yang et al., 2023), whereas GlobalNorm performs min-max normalization across all parameters. Our ablation experiments in Appendix C.5 compare the two strategies alongside the baseline without normalization. The primary evaluation metric is the average accuracy, calculated for each client $i \in \mathcal{C}$ based on their local dataset $\mathcal{D}_i$. The overall metric is the average of these individual accuracies across all clients $\mathcal{C}$.

### 4.2 EXPERIMENTAL RESULTS

**Performance** Table 1 presents a comparative accuracy analysis of our method against eight various baselines methods on four image classification tasks. The results demonstrate that our method consistently achieves superior accuracy under varying levels of non-IID data distributions. Specifically, under $\text{Dir}(0.1)$, `FedOBP` outperforms the second-best method by 0.59% and 8.28% on CIFAR10 and CIFAR100, respectively, and achieves gains of 0.16% on EMNIST. On SVHN, `FedOBP` provides a 0.87% improvement over the second-best FedPer. Even at heterogeneity $\text{Dir}(0.5)$, our method maintains its advantage on all datasets. The superiority of `FedOBP` in tackling the non-IID data distribution issues highlights the advantages of using the `FedOBP` score to determine the parameters that should be personalized. We also conducted tests on ResNet-18, where the distinction between the feature extractor and classifier is less clearly defined. The results can be found in the Appendix C.1.2.

**Convergence** Figure 2 provides the convergence performance of all various algorithms across four datasets with $\alpha = 0.1$. we use a bold red line to highlight `FedOBP`. On CIFAR10, `FedOBP`

Table 1: Average (standard deviation) test accuracy on four datasets. Bold and underlined indicate the best and second-best respectively.

| Dataset | CIFAR10 | | CIFAR100 | | EMNIST | | SVHN | |
|---|---|---|---|---|---|---|---|---|
| Partition | Dir(0.1) | Dir(0.5) | Dir(0.1) | Dir(0.5) | Dir(0.1) | Dir(0.5) | Dir(0.1) | Dir(0.5) |
| Local-Only | 80.87(0.12) | 54.78(0.24) | 37.88(0.50) | 15.76(0.39) | 92.72(0.81) | 85.02(0.23) | 90.15(0.26) | 77.71(0.61) |
| FedAvg | 58.15(0.26) | 63.98(0.29) | 26.18(0.36) | 25.40(0.22) | 82.11(0.12) | 84.02(0.17) | 88.21(0.76) | 90.24(0.66) |
| FedPer | 84.98(0.43) | 65.87(0.67) | 42.09(0.47) | 20.68(0.27) | 94.20(0.02) | 87.55(0.09) | 94.73(0.23) | 89.46(0.65) |
| APFL | 59.37(1.23) | 63.78(0.21) | 26.55(0.47) | 25.10(0.06) | 81.87(0.16) | 83.85(0.15) | 89.51(0.74) | 91.05(0.41) |
| LG-FedAvg | 81.74(0.25) | 57.46(0.87) | 39.08(0.73) | 16.89(0.73) | 93.69(0.09) | 86.32(0.34) | 91.68(0.41) | 81.63(1.10) |
| FedRep | 84.56(0.26) | 63.63(0.49) | 39.35(0.35) | 16.83(0.18) | 94.36(0.22) | 87.38(0.56) | 94.16(0.22) | 86.91(0.43) |
| pFedFDA | 86.43(0.10) | 68.72(0.19) | 41.72(0.45) | 16.71(0.77) | 93.39(0.13) | 86.24(0.21) | 93.95(0.14) | 87.53(0.28) |
| FLUTE | 74.78(0.00) | 48.25(1.19) | 31.61(0.00) | 12.63(1.01) | 80.32(0.00) | 63.87(0.44) | 72.24(0.00) | 43.06(0.89) |
| FedDPA | 81.39(0.02) | 57.37(1.19) | 38.88(0.11) | 16.74(1.27) | 93.73(0.02) | 86.64(0.29) | 91.99(0.01) | 82.83(1.39) |
| FLOCO | 80.77(0.73) | 66.26(0.18) | 27.98(0.63) | 18.25(0.82) | 88.74(0.06) | 85.71(0.20) | 93.37(0.51) | 89.76(0.62) |
| FedALA | 56.94(0.09) | 63.58(0.00) | 26.00(0.09) | 25.69(0.00) | 82.06(0.05) | 83.89(0.00) | 87.61(0.12) | 89.91(0.00) |
| FedSelect | 79.89(0.00) | 53.59(0.00) | 36.33(0.00) | 15.12(0.00) | 92.91(0.00) | 84.57(0.00) | 90.14(0.00) | 77.58(0.00) |
| FedOBP | **87.02(0.17)** | **70.22(0.16)** | **50.37(0.16)** | **27.05(0.30)** | **94.36(0.04)** | **88.82(0.05)** | **95.60(0.03)** | **91.75(0.47)** |

achieves the highest accuracy and comparable convergence. pFedFDA and FedPer also show good convergence reaching above 85%. Other methods, such as APFL, FLUTE and FedAL, converge slowly, with final accuracies below 75%. On CIFAR100 and EMNIST, FedOBP displays the fastest convergence and highest final accuracy compared to all alternative solutions. Additionally, FedOBP demonstrates comparable convergence on SVHN while achieving best final accuracy. Figure 3 shows the convergence results with $\alpha = 0.5$. FedOBP shows best accuracy with comparable convergence, achieving optimal results at approximately 200 epochs for CIFAR10 and 200 epochs for CIFAR100, respectively. FedOBP demonstrates fastest convergence, achieving the first position of accuracy in SVHN and shows fast convergence on EMNIST. These results indicate that FedOBP achieves competitive convergence across four datasets with varying levels of data heterogeneity.

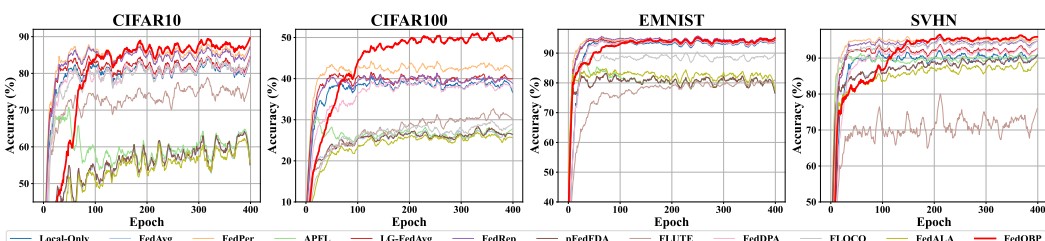

Figure 2: Convergence comparison of FedOBP and ten other solutions with $\alpha = 0.1$ on the 4-layer CNN model across four datasets.

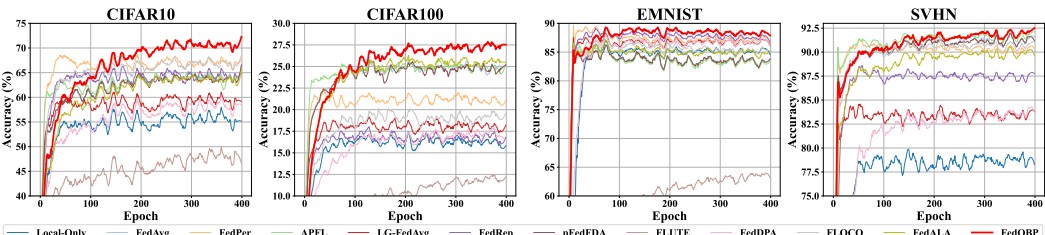

Figure 3: Convergence comparison of FedOBP and ten other solutions with $\alpha = 0.5$ on the 4-layer CNN model across four datasets.

**Importance Scores** We further analyze the performance of three types of scores (Gradient $I_G(\cdot)$, Fisher $I_F(\cdot)$, and FedOBP $I_O(\cdot)$) with different quantile $q$ settings ranging from 0.0 to 1.0, where a larger quantile indicates a smaller proportion $(1 - q)$ of personalized parameters. This analysis employs a 4-layer CNN model on both the CIFAR-10 and CIFAR-100 datasets. The total number of FL rounds is set to 200, with the default parameter value $\gamma = 0.1$.

Across all datasets, as the quantile $q$ increases, leading to fewer personalized parameters $\boldsymbol{u}_i$ and more global parameters $\boldsymbol{v}_i$, all three scores initially rise and then decline. Each score demonstrates a clear phase transition upon achieving peak performance at varying quantile values across different datasets. As shown in Figure 4, for CIFAR10, $I_G(\cdot)$ peaks at quantile $q = 0.1$, $I_F(\cdot)$ at $q = 0.7$, and $I_O(\cdot)$ at

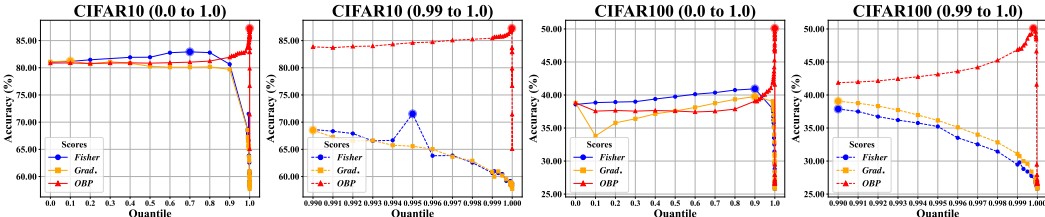

Figure 4: Comparison of three types of scores, including Gradient $I_G(\cdot)$, Fisher $I_F(\cdot)$, and OBP $I_O(\cdot)$, on CIFAR10 and CIFAR100.

$q = 0.9999$. These results indicate that to achieve optimal performance, $I_G(\cdot)$ requires approximately $790,684$ personalized parameters, $I_F(\cdot)$ needs $263,561$, while $I_O(\cdot)$ requires only $87$. If we increase the FL rounds to $400$, the number of personalized parameters required for $I_O(\cdot)$ to achieve optimal results will further decrease to $18$ as shown in Appendix B.3. On CIFAR100, $I_G(\cdot)$ peaks at quantile $q = 0.9$ ($87,853$), $I_F(\cdot)$ at quantile $q = 0.9$ ($87,853$), and $I_O(\cdot)$ at $q = 0.9998$ ($185$). These results demonstrate that $I_O(\cdot)$ score can accurately identify the necessary personalized parameters. This phenomenon is also observed in the other datasets in Appendix C.3.

**Personalized Parameter Distribution**   Figure 5 illustrates the proportion of personalized parameters across various layers of the 4-layer CNN model during training over $450$ epochs for four different datasets. Across all datasets, a consistent trend is observed where the proportion of personalized parameters in the first convolutional layer (conv1) gradually decreases, while the proportion in the classifier layer increases over time. On CIFAR10 and CIFAR100, this distribution stabilizes between $300$ and $450$ epochs, with the classifier layer reaching $0.9$ to $1.0$ and conv1 stabilizing at $0.0$ to $0.1$. In EMNIST, stability is reached around $250$ to $450$ epochs, with the classifier layer at $0.7$ to $0.8$ and conv1 at $0.2$ to $0.3$. For SHVN, stability occurs between $300$ to $450$ epochs, with the classifier layer at $0.7$ to $0.9$ and conv1 at $0.1$ to $0.3$.

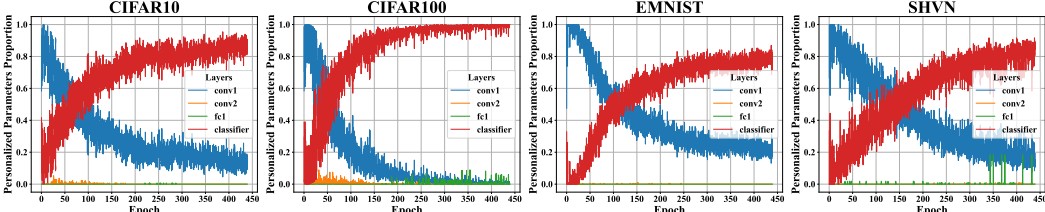

Figure 5: Personalized parameters distribution across layers varies with FL epochs using the 4-layer CNN model on four datasets.

Many layer-wise model decoupling methods (Collins et al., 2021; OH et al., 2022; Mclaughlin & Su, 2024), based on CKA theory (Hinton et al., 2015), designate the final classifier layer as the personalization layer. However, these methods typically treat all parameters in the classifier layer as personalized, representing a simplified case compared to our `FedOBP`. In contrast, `FedOBP` selectively identifies which parameters to personalize and has automatically discovered that only a small subset of parameters in the classifier layer is sufficient for effective personalization. To some extent, our results on personalized parameter selection align with the "last-layer-as-personalization" insight from CKA-based methods, confirming that `FedOBP` score effectively identifies personalized parameters.

## 5 CONCLUSION

In this paper, we address the challenge in PFL of identifying which parameters should be personalized to effectively handle data heterogeneity across clients. We propose a parameter decoupling algorithm that incorporates a quantile-based thresholding mechanism. In addition, we introduce an element-wise importance score, referred to as Federated Optimal Brain Personalization (`FedOBP`). This score is building on OBD pruning theory and utilizes a federated approximation of the first order derivative in the Taylor series expansion to assess the significance of each local parameter in relation to the global dataset $\mathcal{D}$. Finally, we evaluate `FedOBP` on various datasets with different heterogeneity settings and show that it outperforms baseline methods. Future research directions include developing adaptive thresholding methods that go beyond static quantile-based approaches with fixed thresholds ($\tau$). Additionally, exploring soft combinations of local and global model parameters represents another promising direction.

ETHICS STATEMENT

The authors of this work have read and commit to adhering to the Code of Ethics. Our research proposes a personalized Federated Learning framework inspired by model pruning and, to the best of our knowledge, does not present any direct ethical concerns. The work does not involve the use of personally identifiable information, sensitive human-subject data, or applications with immediate potential for societal harm.

REPRODUCIBILITY STATEMENT

We provide the complete source code in the supplementary materials. Further details on the experimental setup, including hyperparameters, datasets, model architecture and computing environment, are documented in the Appendix.

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

# Appendix

## A DISCUSSION

### A.1 LIMITATIONS

The proposed `FedOBP` has several limitations that suggest promising directions for future research.

First, the threshold parameter $\tau$, which governs optimal quantiling, significantly impacts performance. While selecting a high threshold (e.g., 0.9995) to focus on a small subset often works well across diverse datasets, identifying the optimal $\tau$ still requires tuning. Developing adaptive methods to learn or adjust $\tau$ based on group-level signals could enhance generalization and reduce manual calibration.

Second, our use of a CNN model reflects a common practice in federated learning, where more complex architectures like ResNet require batch normalization, which is known to underperform in non-IID federated settings (Diao et al., 2021; Li et al., 2021). As a result, many prior works rely on simpler networks for demonstration, and evaluating `FedOBP` with more advanced architectures remains an important direction.

### A.2 USE OF LARGE LANGUAGE MODELS

In this work, we used large language models (LLMs) to assist with manuscript editing. LLMs were used to help polish the language of the manuscript. This includes surface-level edits such as improving clarity, grammar, and conciseness of English expressions. All technical content, algorithmic designs, and empirical results were authored and validated by the authors. No part of the scientific contributions was generated by or delegated to an LLM.

## B EXPERIMENTAL SETUP

### B.1 COMPUTING ENVIRONMENT

We implement our experiments using PyTorch 2.1.0+cu121 and run all experiments on Microsoft Windows 11 Professional Edition. Our system configuration includes a 13th Gen Intel(R) Core(TM) i9-13900K CPU @ 3.00GHz with 24 cores and 32 logical processors, combined with an NVIDIA GeForce RTX 4090 GPU. The system is built on a Micro-Star International Co., Ltd. MS-7D25 model (PRO Z690-A WIFI DDR4) with 64.0 GB of RAM.

### B.2 DATASETS DETAILS

Table 2: Statistical information of used datasets on clients.

| Dataset | Samples | Classes | Description | Resolution | Year |
|---|---|---|---|---|---|
| CIFAR10/100 | 60,000 | 10/100 | Images in 10/100 classes including airplanes, cars, birds, etc. | $32 \times 32$ | 2009 |
| EMNIST | 805,263 | 62 | Extended MNIST with letters and digits | $28 \times 28$ | 2017 |
| FMNIST | 70,000 | 10 | Fashion item images | $28 \times 28$ | 2017 |
| MNIST | 70000 | 10 | Handwritten digits | $28 \times 28$ | 1998 |
| MEDMNISTA/C | 58,850/23,600 | 11 | Biomedical images on abdominal CT. | $28 \times 28$ | 2019 |
| SVHN | 600,000 | 10 | Street view house numbers | $32 \times 32$ | 2011 |

The fundamental details of the datasets used in our experiments are presented in Table 2. Each dataset is partitioned into 100 subsets and assigned to 100 clients to simulate the non-IID scenario. The data distribution remains consistent throughout the experiments, ensuring a uniform basis for evaluating the effectiveness of various methods.

### B.3 MODEL ARCHITECTURE

A four-layer neural network is utilized as the backbone model in our experiments. The detailed architecture is presented in Table 3, with layer abbreviations following the PyTorch style. In Table 5,

Table 3: The 4-layer CNNs. Conv2D consists of a 2D convolution layer, ReLU activation, and MaxPool2D layer executed sequentially. The *in, out, kernel* represent the input channel, output channel, and kernel size, respectively; MaxPool2D is a max pooling layer for 2D input; Flatten reshapes input from 2D to 1D; FC is a fully connected layer, where *out* indicates the number of output features.

| Layer | Operation | Parameters |
|---|---|---|
| Conv2D (conv1) | 2D Convolution | INPUT_CHANNELS $\times 32 \times 5 \times 5 + 32$ |
| ReLU (activation1) | Activation | - |
| MaxPool2D (pool1) | Max Pooling | - |
| Conv2D (conv2) | 2D Convolution | $32 \times 5 \times 5 + 64$ |
| ReLU (activation2) | Activation | - |
| MaxPool2D (pool2) | Max Pooling | - |
| Flatten (flatten) | Reshape | - |
| FC (fc1) | Fully Connected | $64 \times 4 \times 4 \times 512 + 512$ |
| ReLU (activation3) | Activation | - |
| Classifier | FC (*out=NUM_CLASSES*) | $512 \times$ NUM_CLASSES + NUM_CLASSES |

Table 4: ResNet-18 Architecture. Each block consists of a series of 2D convolution layers, followed by ReLU activation and batch normalization. The *in, out, kernel* represent the input channel, output channel, and kernel size, respectively; FC is a fully connected layer, where *out* indicates the number of output features.

| Layer | Operation | Parameters |
|---|---|---|
| Conv2D (conv1) | 2D Convolution | $3 \times 64 \times 7 \times 7 + 64$ |
| ReLU (activation1) | Activation | - |
| BatchNorm (bn1) | Batch Normalization | - |
| MaxPool2D (pool1) | Max Pooling | |
| Residual Block (block1) | 2 Conv2D + ReLU + BatchNorm | $64 \times 64 \times 3 \times 3 + 64, 64 \times 64 \times 3 \times 3 + 64$ |
| Residual Block (block2) | 2 Conv2D + ReLU + BatchNorm | $64 \times 128 \times 3 \times 3 + 128, 128 \times 128 \times 3 \times 3 + 128$ |
| Residual Block (block3) | 2 Conv2D + ReLU + BatchNorm | $128 \times 256 \times 3 \times 3 + 256, 256 \times 256 \times 3 \times 3 + 256$ |
| Residual Block (block4) | 2 Conv2D + ReLU + BatchNorm | $256 \times 512 \times 3 \times 3 + 512, 512 \times 512 \times 3 \times 3 + 512$ |
| FC (fc1) | Fully Connected | $512 \times 1000 + 1000$ |
| Classifier | FC (*out=num_classes*) | $1000 \times$ NUM_CLASSES + NUM_CLASSES |

we present the `FedOBP` quantile settings for the eight datasets (MNIST, FMNIST, MEDMNISTA and MEDMNISTC) under two heterogeneity configurations ($\alpha \in \{0.1, 0.5\}$).

### B.4 Hyperparameter settings for comparative methods

We present a detailed overview of the hyperparameter settings for all the comparison methods discussed above. Most baseline hyperparameters are consistent with the values reported in their respective original papers.

- FedRep (Collins et al., 2021) We set the epoch for training feature extractor part to 1.
- FedDPA (Yang et al., 2023) We set the fisher threshold $\tau_F = 0.4$.

## C  Experimental Results

### C.1  Performance

#### C.1.1  Performance for AvgCNN (4-layer CNNs)

The results summarized in Table 6 provide a comprehensive evaluation of the average test accuracy across multiple datasets, comparing our method, `FedOBP`, with various SOTA approaches and a Local-Only baseline under different non-IID data distributions.

In the MNIST dataset, `FedOBP` achieves the highest accuracy of 99.34% under the Dir(0.1) partition, surpassing the second-best method, FedPer, by 0.31%. Under the Dir(0.5) partition, `FedOBP` maintains a competitive accuracy of 98.97%, closely following APFL. For the FMNIST dataset, `FedOBP` again leads with an accuracy of 96.89% under Dir(0.1), showing an improvement of 0.65%

Table 5: Quantile($q$) thresholds and corresponding number of personalized parameters for Dir(0.1) and Dir(0.5) across four datasets.

| Datasets | Dir(0.1) | | Dir(0.5) | |
|---|---|---|---|---|
| | $q$ | Number | $q$ | Number |
| CIFAR10 | 0.99998 | 18 | 0.99979 | 185 |
| CIFAR100 | 0.9998 | 185 | 0.99992 | 74 |
| EMNIST | 0.995 | 3,044 | 0.9991 | 548 |
| SVHN | 0.99997 | 27 | 0.9998 | 176 |
| MNIST | 0.9998 | 117 | 0.99993 | 41 |
| FMNIST | 0.99993 | 41 | 0.9999 | 59 |
| MEDMNISTA | 0.9999 | 53 | 0.99995 | 30 |
| MEDMNISTC | 0.9997 | 175 | 0.99995 | 30 |

over the next best method, FedPer. In the Dir(0.5) partition, it also achieves 93.11%, marking a 1.73% increase compared to FLOCO. In the MEDMNISTA dataset, FedOBP records an accuracy of 67.31% under Dir(0.1), with only a 0.16% gap from the optimal solution. However, under Dir(0.5), it achieves 41.22%, which ties with pFedFDA as the best-performing solution. Lastly, for the MEDMNISTC dataset, FedOBP achieves 66.84% under Dir(0.1) and 41.36% under Dir(0.5), both of which are competitive but slightly below the best performances.

Overall, the results indicate that FedOBP consistently delivers strong performance across various datasets and non-IID distributions, demonstrating its effectiveness in addressing the challenges posed by heterogeneous data.

Table 6: Average (standard deviation) test accuracy on multiple datasets. Bold and underlined indicate the best and second-best respectively.

| Dataset | MNIST | | FMNIST | | MEDMNISTA | | MEDMNISTC | |
|---|---|---|---|---|---|---|---|---|
| Partition | Dir(0.1) | Dir(0.5) | Dir(0.1) | Dir(0.5) | Dir(0.1) | Dir(0.5) | Dir(0.1) | Dir(0.5) |
| Local-Only | 97.49(0.05) | 94.44(0.13) | 94.93(0.08) | 86.49(0.24) | 67.22(0.05) | 41.08(0.07) | 66.92(0.07) | 41.21(0.15) |
| FedAvg | 98.58(0.13) | 98.89(0.08) | 87.29(0.45) | 90.37(0.22) | 18.10(0.00) | 18.28(0.55) | 22.28(0.00) | 22.33(0.11) |
| FedPer | 99.03(0.06) | 98.10(0.22) | 96.24(0.19) | 91.06(0.38) | 67.44(0.03) | 41.10(0.05) | 66.83(0.02) | 41.35(0.06) |
| APFL | 98.98(0.18) | **99.04(0.09)** | 89.37(0.51) | 91.00(0.39) | 58.50(0.97) | 35.89(1.78) | 56.06(0.74) | 38.81(0.19) |
| LG-FedAvg | 97.82(0.13) | 95.46(0.45) | 95.24(0.07) | 87.36(0.30) | 67.23(0.07) | 41.10(0.13) | **66.97(0.01)** | 41.34(0.03) |
| FedRep | 98.59(0.10) | 97.16(0.20) | 95.80(0.20) | 89.85(0.46) | 67.09(0.12) | 41.08(0.08) | 66.35(0.16) | 41.18(0.05) |
| pFedFDA | 98.78(0.09) | 97.80(0.22) | 96.09(0.15) | 91.10(0.48) | **67.47(0.00)** | 41.22(0.01) | 66.79(0.00) | **41.45(0.00)** |
| FLUTE | 91.13(0.00) | 80.48(0.00) | 88.56(0.00) | 72.83(0.00) | **67.47(0.00)** | 41.14(0.00) | 66.79(0.00) | 41.27(0.00) |
| FedDPA | 98.04(0.00) | 95.92(0.52) | 95.06(0.00) | 88.01(0.51) | 67.02(0.10) | 40.34(0.97) | 66.66(0.05) | 40.94(0.34) |
| FLOCO | 98.80(0.17) | 98.50(0.10) | 95.06(0.58) | 91.38(0.54) | 59.47(0.23) | 33.40(0.44) | 60.66(1.26) | 34.78(0.89) |
| FedALA | 98.58(0.04) | 98.88(0.04) | 86.98(0.04) | 90.18(0.00) | 18.10(0.00) | 17.96(0.00) | 9.03(0.00) | 22.27(0.00) |
| FedSelect | 97.46(0.00) | 94.44(0.00) | 93.83(0.00) | 86.21(0.00) | 67.13(0.00) | 40.96(0.00) | 66.62(0.00) | 41.32(0.00) |
| FedOBP | **99.34(0.02)** | 98.97(0.04) | **96.89(0.03)** | **93.11(0.04)** | 67.31(0.00) | **41.22(0.13)** | 66.84(0.06) | 41.36(0.09) |

## C.1.2 PERFORMANCE FOR RESNET-18

Table 7: Average test accuracy on multiple datasets under ResNet-18. Bold and underlined indicate the best and second-best respectively. The personalized parameter ratio selected by FedOBP is shown in parentheses.

| Method | CIFAR10 | CIFAR100 | EMNIST | FMNIST | MNIST | SVHN |
|---|---|---|---|---|---|---|
| FedAvg | 69.33 | 45.33 | 82.07 | 89.93 | 99.19 | 89.45 |
| Local-Only | 86.25 | 50.71 | 93.17 | 95.57 | 98.13 | 91.62 |
| APFL | 70.60 | 45.70 | 81.78 | 90.24 | 99.23 | 90.05 |
| FedALA | 69.95 | 45.30 | 82.22 | 90.02 | 99.19 | 89.33 |
| FedDPA | 86.27 | 51.15 | 93.53 | 95.75 | 98.23 | 91.94 |
| FedPer | **90.36** | 64.18 | 94.18 | **97.08** | **99.63** | **95.97** |
| FedRep | 90.10 | 58.18 | 94.42 | 96.60 | 99.42 | 95.68 |
| FedSelect | 85.70 | 50.02 | 92.91 | 95.60 | 98.11 | 91.40 |
| FLUTE | 71.73 | 35.81 | 80.29 | 92.05 | 93.32 | 71.02 |
| LG-FedAvg | 87.01 | 53.32 | 93.66 | 95.96 | 98.60 | 93.33 |
| FedOBP | 90.04 (0.14%) | **64.58** (0.15%) | **95.31** (0.40%) | 96.93 (0.18%) | 99.48 (0.06%) | 95.57 (0.16%) |

We evaluated the performance of our method on ResNet-18, with architectural details provided in 4. The experiment used a heterogeneity parameter of $\alpha = 0.1$. As shown in Table 7, FedOBP achieves impressive results even with minimal parameter personalization, particularly on CIFAR100 and EMNIST, where it outperforms all baselines with performance scores of 64.58% and 95.31%, respectively. Notably, on the MNIST dataset, our method achieves 99.48% performance by personalizing just 0.06% of the model parameters. These results demonstrate the adaptability of our approach across different architectures, performing excellently not only on simpler models like AvgCNN (4-layer CNNs) but also on more complex ones such as ResNet-18.

## C.2  CONVERGENCE

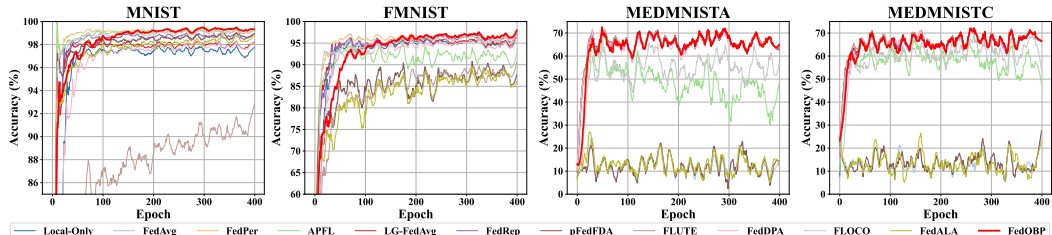

Figure 6: Convergence comparison of FedOBP and ten other solutions with $\alpha = 0.1$ on the 4-layer CNN model across four datasets.

Figures 6 and 7 illustrate the convergence performance of eleven algorithms across MNIST, FMNIST, MEDMNISTA, and MEDMNISTC under $\alpha = 0.1$ and $\alpha = 0.5$, respectively. On MNIST and FMNIST, FedOBP outperforms or is slightly inferior to other methods in terms of accuracy and convergence rates. However, on MEDMNISTA and MEDMNISTC, it slightly lags behind all other methods in accuracy but demonstrates faster convergence. Among model decoupling approaches, LG-FedAvg and pFedFDA also demonstrate competitive performance, surpassing 67% and 66% accuracy on MEDMNISTA and MEDMNISTC under $\alpha = 0.1$, respectively. Overall, FedOBP proves to be a robust solution across different datasets and heterogeneity levels.

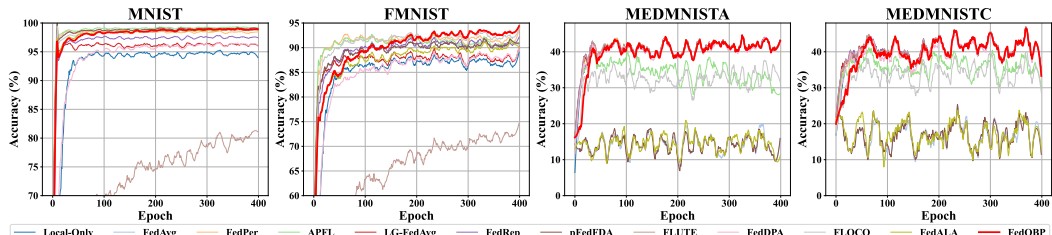

Figure 7: Convergence comparison of FedOBP and ten other solutions with $\alpha = 0.5$ on the 4-layer CNN model across four datasets.

## C.3  IMPORTANCE SCORES

Across all datasets, as the quantile $q$ increases, resulting in a reduction of personalized parameters $\boldsymbol{u}_i$ and an increase in global parameters $\boldsymbol{v}_i$, the accuracy scores for all three scores initially rise before experiencing a decline. To achieve optimal performance, $I_G(\cdot)$ requires approximately 60% personalized parameters, $I_F(\cdot)$ needs 30%, while $I_O(\cdot)$ only requires less than 0.1% on MNIST. For FMNIST, $I_G(\cdot)$ peaks at $q = 0.1$, $I_F(\cdot)$ reaches its maximum at quantile $q = 0.4$, and $I_O(\cdot)$ peaks at $q = 0.99993$. This suggests that the $I_O(\cdot)$ score effectively identifies the necessary few personalized parameters. For MEDMNISTA and MEDMNISTC, the OBP score shows stable performance across various quantile ranges compared to the gradient and Fisher-based scores.

## C.4  PERSONALIZED PARAMETER DISTRIBUTION

Figure 10 illustrates the proportion of personalized parameters across various layers of the 4-layer CNN model during training over 450 epochs across four different datasets. For all datasets, there is a clear trend where the proportion of personalized parameters located at the conv1 decreases over time, while the proportion at the classifier layer increases. On MNIST stability is achieved between 350 to 450 epochs, with the classifier layer reaching 0.4 to 0.8 and conv1 stabilizing at 0.2 to 0.6. FMNIST stability occurs around 250 to 450 epochs, with the classifier layer at 0.5 to 0.6 and conv1

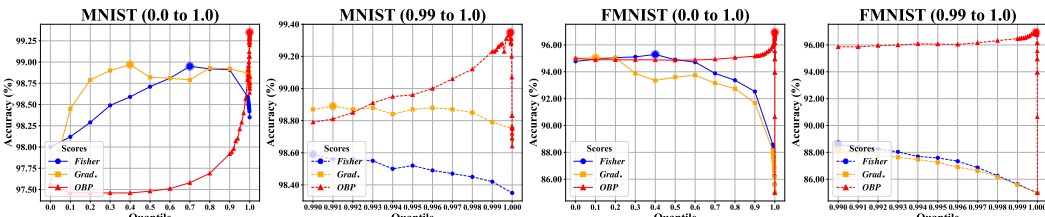

Figure 8: Comparison of three types of scores, including Gradient $I_G(\cdot)$, Fisher $I_F(\cdot)$, and OBP $I_O(\cdot)$, on MNIST and FMNIST.

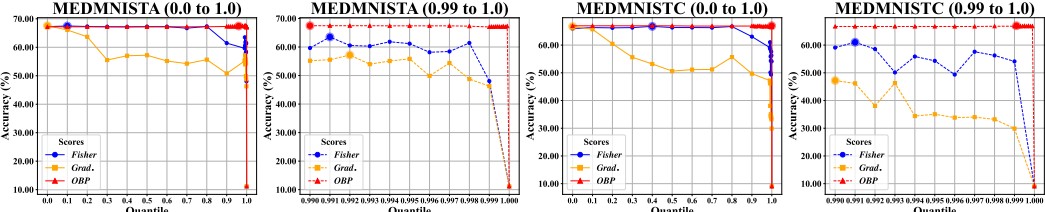

Figure 9: Comparison of three types of scores, including Gradient $I_G(\cdot)$, Fisher $I_F(\cdot)$, and OBP $I_O(\cdot)$, on MEDMNISTA and MEDMNISTC.

at 0.4 to 0.5. MEDMNISTA and MEDMNISTC stability is observed from 50 to 450 epochs, with the classifier layer at 0.8 to 1.0 and conv1 at 0.0 to 0.2.

## C.5 ABLATION STUDIES

The ablation studies evaluate the performance by implementing three normalization techniques for our `FedOBP` score $I_O(\cdot)$ including NoNorm, LayerNorm, and GlobalNorm across the four datasets. NoNorm apply raw `FedOBP` scores for personalized parameter selection. LayerNorm normalizes the `FedOBP` scores in layer-wise which is adapt in (Yang et al., 2023). GlobalNorm computes normalization statistics across the entire model. We explore two additional variants of GlobalNorm including GlobalNorm without CLS (w/o CLS), GlobalNorm with CLS, where the CLS variant selects personalized parameters from the classifier layer.

The ablation studies in Table 8 show that NoNorm achieves the highest accuracy in FMNIST (96.82%) and MEDMNISTC (66.60%), while GlobalNorm without CLS yields the best performance in MNIST (99.40%) and MEDMNISTA (67.47%). LayerNorm consistently underperforms across all datasets, indicating that GlobalNorm are more effective normalization techniques for the evaluated tasks.

Table 8: Ablation experiment comparing NoNorm, LayerNorm, and GlobalNorm on four datasets. Additionally, the ablation study examines GlobalNorm w/o CLS and with CLS.

| Dataset | NoNorm | LayerNorm | GlobalNorm w/o CLS | GlobalNorm with CLS |
|---|---|---|---|---|
| CIFAR10 | 87.36 | 85.00 | 87.36 | **87.37** |
| CIFAR100 | **45.98** | 41.83 | **45.98** | 44.89 |
| EMNIST | **94.92** | 94.14 | **94.92** | 94.78 |
| SHVN | **95.88** | 93.34 | **95.88** | **95.88** |
| MNIST | 99.38 | 99.27 | 99.38 | **99.39** |
| FMNIST | **96.82** | 96.31 | **96.82** | 96.81 |
| MEDMNISTA | **67.47** | 67.39 | **67.47** | **67.47** |
| MEDMNISTC | **66.60** | 65.17 | **66.60** | 66.38 |

Figure 11 shows the ablation experiment on MNIST, FMNIST, MEDMNISTA and MEDMNISTC datasets. NoNorm and GlobalNorm (NA) consistently achieved high accuracy with quick convergence across all datasets. LayerNorm underperforms, especially in MNIST, FMNIST, and MEDMNISTC, with significantly lower accuracy in MNIST and FMNIST. Among GlobalNorm variations, the configuration without CLS delivers the highest accuracy.

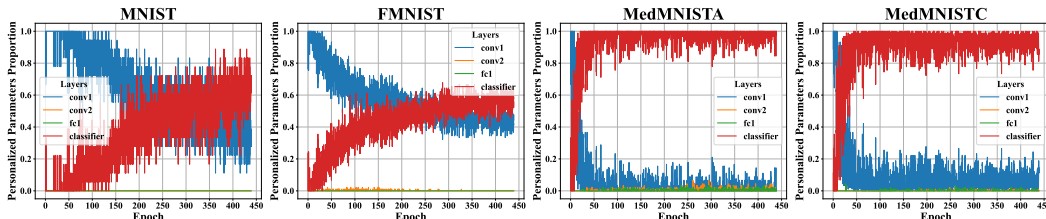

Figure 10: Personalized parameters distribution across layers varies with FL epochs using the 4-layer CNN model on four datasets.

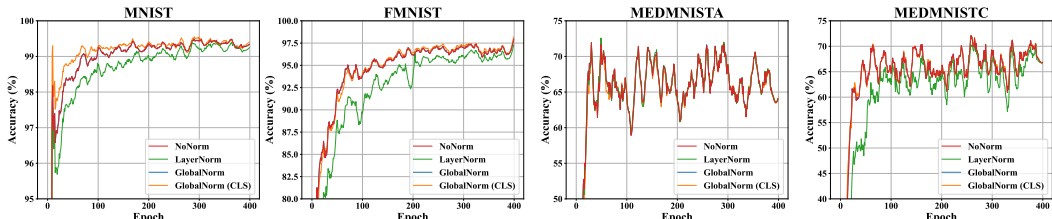

Figure 11: Ablation experiment comparing NoNorm, LayerNorm, and GlobalNorm on MNIST, FMNIST, MEDMNISTA, and MEDMNISTC datasets. GlobalNorm has two variants, one with CLS and one without (w/o CLS).

As demonstrated in Figure 12, both NoNorm and GlobalNorm consistently outperformed LayerNorm in terms of accuracy. In the CIFAR10 dataset, NoNorm and GlobalNorm (without CLS) achieve an accuracy of 87.52%, while LayerNorm records 85.00%. In CIFAR100, NoNorm and GlobalNorm again lead with 45.98%, whereas LayerNorm drops to 41.83%. GlobalNorm configurations perform similarly to NoNorm. For the EMNIST dataset, NoNorm and GlobalNorm (NA) achieve an accuracy of 94.92%, outperforming LayerNorm, which achieves 94.14%. The SHVN dataset shows similar results to CIFAR10 and CIFAR100. Notably, LayerNorm tends to select at least one personalized parameter per layer, whereas NoNorm and GlobalNorm primarily select personalized parameters from the classifier layer, indicating that LayerNorm may degrade the performance of `FedOBP` score.

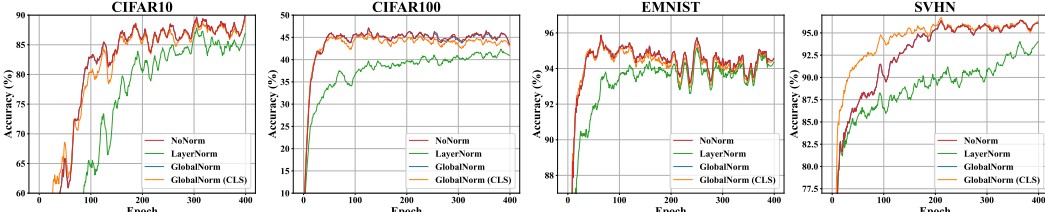

Figure 12: Ablation experiment comparing NoNorm, LayerNorm, and GlobalNorm (including w/o CLS and with CLS) on CIFAR10, CIFAR100, EMNIST, and SVHN datasets

Compared with selecting personalized parameters globally (w/o CLS), selecting personalized parameters only from the classifier layer (CLS) produces comparable results in CIFAR10 and SVHN datastes and slightly worse results of 44.89 % in CIFAR100 datasets and 94.78% in EMNIST dataset. However, constraining personalized parameter selection to the classifier layer reduces the algorithm's complexity and may be a promising direction for future works.

Overall, the ablation experiment results demonstrate that LayerNorm negatively affects the performance of the `FedOBP` score, while GlobalNorm combined with CLS can slightly degrade the performance. Notably, even without normalization, the `FedOBP` performance remains robust, showcasing the scalability of our scoring method.

