# OpenReview forum: "FedOBP: Federated Optimal Brain Personalization with Few Personalized Parameters"
_ICLR.cc/2026/Conference — ICLR 2026 Conference Withdrawn Submission_

### Official Review · Reviewer_zizm · 2025-10-16

**Soundness:** 3
**Presentation:** 2
**Contribution:** 2
**Rating:** 4
**Confidence:** 4

**Summary:**

This paper proposes FedOBP, a PFL method that uses an importance score and quantile-based thresholding to select which parameters to personalize. It achieves state-of-the-art performance across diverse datasets while personalizing only a small subset of parameters.

**Strengths:**

- The proposed method is conceptually reasonable and makes intuitive sense, even though it lacks a rigorous theoretical justification.
- The paper shows comparisons with many baselines, including recently published baselines.

**Weaknesses:**

- Small-scale experiments:
    - The paper only evaluates on CIFAR, MNIST, and SVHN, which are relatively small-scale benchmarks. Including results on large-scale datasets (e.g., ImageNet) is necessary to demonstrate broader applicability.
    - Additionally, the experiments use only 4-layer CNNs and ResNet-18, which are simple and small-scale models. Evaluating on larger-scale architectures (e.g., ViT-based models) is needed to assess the scalability of the proposed method.

- Conflicting arguments: In Fig. 2 and Fig. 3, while the proposed method achieves the highest convergence accuracy, it converges more slowly than other methods. However, the results section claims faster convergence, which is contradictory. The paper should clarify the reason for the slower convergence speed.

- Marginal improvements in larger-size architecture: Although the method achieves the highest performance with a 4-layer CNN, in ResNet-18 experiments (Table 7), it shows lower accuracy than several baselines, raising concerns about its generalizability and scalability.

- Errors and typos:
    - Line 303: “Eq.” and “Equation.” are duplicated.
    - Line 43: Citations for PSPFL and FedSelect should use \citep instead of \citet.
    - Table 1 (EMNIST column): Both FedOBP and FedRep achieve 94.36, but only FedOBP is marked in bold. This misreporting reduces trust in the results.

**Questions:**

NA

---

### Official Review · Reviewer_yEGg · 2025-10-29

**Soundness:** 3
**Presentation:** 1
**Contribution:** 2
**Rating:** 4
**Confidence:** 3

**Summary:**

This paper addresses the problem of personalized federated learning (PFL) in the presence of data heterogeneity across clients. To determine which parameters should be personalized (local) or shared (global), the authors propose Federated Optimal Brain Personalization (FedOBP), an element-wise importance score that quantifies the discrepancy between local and global parameters. By decoupling local and global parameters, the method aggregates the global ones across clients while keeping the personalized ones locally. Experimental results demonstrate that the proposed federated learning algorithm with FedOBP achieves state-of-the-art performance while requiring only a small number of personalized parameters.

**Strengths:**

- The proposed parameter importance score is simple and seems to be well-principled.
- The proposed algorithm outperforms existing PFL methods.

**Weaknesses:**

- **Clarity and notation issues.** Overall, this paper is difficult to follow, mainly due to confusing notation and unclear algorithmic descriptions.
  - Is $\tilde{\theta}$ the initialization parameter? Then, is $\theta_i^t$ the updated version after local training? At this point, are both local and global parameters updated simultaneously during training each client? It should be clearly described which parameters are initialized at each communication round, how local updates modify them, and which parameters are aggregated before proceeding to the next round.
  - In L205-206, the notations are unclear because $I_O(\theta_i^{t-1};\mathcal{D})$ is a set of importance scores, not a scalar value.
  - I think the global objective cannot be explicitly expressed as Equation (3); it should rather be formulated as a bi-level optimization problem.
- **Insufficient experiments**
  - The paper does not clearly explain how data heterogeneity is controlled during dataset construction. Given $\alpha$, how to split data across clients? Does heterogeneity mean non-overlapping classes or merely different samples per client? A detailed explanation is needed.
  - If heterogeneity only refers to sample partitioning, can it truly be regarded as data heterogeneity? In federated settings, clients often have different datasets (e.g., clients 1-5 use CIFAR, clients 6-10 use SVHN), and such scenarios should be tested.
  - Experiments are conducted only on small-scale datasets such as CIFAR, EMNIST, MNIST, and SVHN. The scalability of the method to large-scale datasets should be evaluated.
  - In realistic scenarios, clients typically start from large-scale pretrained models and apply parameter-efficient tuning methods such as LoRA. It is unclear whether the proposed approach would still perform well in such settings.
- **Methodological limitations**
  - The method assumes identical model architectures across clients. It is unclear how the approach would generalize when clients use heterogeneous models.
  - The algorithm requires transferring all model parameters, which can lead to significant communication costs as model size increases.
  - The approach may not be applicable when full gradient sharing is restricted due to privacy concerns. How could the method be adapted under such constraints?
- **Minor Comments**
  - Figure 4 should use a logarithmic scale on the x-axis for better visualization.

**Questions:**

See the above weaknesses part.

---

### Official Review · Reviewer_PPcn · 2025-10-30

**Soundness:** 2
**Presentation:** 3
**Contribution:** 3
**Rating:** 6
**Confidence:** 3

**Summary:**

This paper focuses on Personalized Federated Learning (pFL) for image classification tasks, aiming to address the challenge of balancing personalization (adaptation to local non-IID data) and generalization (utilization of global shared knowledge). The core idea is to leverage the Optimal Brain Personalization (OBP) framework—derived from Optimal Brain Damage (OBD) pruning theory—to screen the importance of individual neurons (model parameters). Notably, the paper demonstrates that only a tiny subset of high-importance parameters needs to be personalized to achieve the desired balance between personalization and generalization. For computational efficiency, the OBP score is approximated using the product of gradients and the difference between global model parameters and local model parameters, avoiding complex second-order calculations and reducing overhead.

**Strengths:**

- High practical value: The paper proposes a simple, scalable, and communication-friendly criterion for personalized parameter selection. The OBP score calculation relies on gradients and global-local parameter differences—information already available in standard FL workflows—eliminating the need for additional data collection or complex computations (e.g., Fisher information estimation in FedDPA). This makes the method easy to integrate into existing FL pipelines.
- Efficient OBD migration: By adapting OBD pruning theory to pFL, the method leverages federated learning’s inherent global-local parameter dynamics without introducing significant extra computational or communication overhead. Unlike gradient-based methods (e.g., FedSelect) that suffer from delays, or Fisher-based methods (e.g., FedDPA) that require extra gradient computations, FedOBP balances efficiency and effectiveness.
- Strong generalization guarantee: The tiny subset of personalized parameters (often <0.5%, e.g., 0.06% for MNIST) ensures that local models retain most globally shared knowledge. This design mitigates the risk of overfitting to local data and maintains strong generalization across clients, which is a key advantage over prior methods that require 30–60% personalized parameters.
Empirically consistent parameter analysis: The experimental observation that personalized parameters are concentrated in the classifier layer aligns with established empirical rules (e.g., CKA-based layer-wise decoupling). This consistency validates the OBP score’s ability to identify semantically meaningful parameters, enhancing trust in the method’s intuitive logic.

**Weaknesses:**

- Limited justification for theoretical approximations: The OBP score’s approximation (using first-order gradient-parameter difference products) is not rigorously validated. Under complex FL scenarios—such as multi-step local updates (more than 5 local epochs) or biased aggregation (e.g., weighted by non-uniform client sample sizes)—the approximation error could grow significantly. The paper also only provides empirical reasons (first-order terms dominate in FL training) for neglecting OBD’s second-order term, without theoretical bounds on its impact on parameter importance ranking.
- Lack of adaptive threshold q: The quantile threshold q is critical to performance (it determines the size of the personalized parameter subset), but the paper only provides dataset-specific empirical values (e.g., q=0.99998 for CIFAR10). When extending to complex setups—such as advanced aggregation strategies (e.g., Scaffold, FedProx) or deep architectures (e.g., ResNet-50, Transformers)—static q may fail to adapt to varying data heterogeneity or model dynamics. An adaptive q (e.g., adjusted based on local loss or global accuracy) would significantly improve generalizability.
- Narrow experimental scope: The method is claimed to be architecture- and task-agnostic, but experiments are limited to simple 4-layer CNNs and image classification tasks (no tests on Transformers, vision transformers (ViTs), or non-classification tasks like image segmentation). This narrow scope raises doubts about its applicability to more complex scenarios (e.g., medical image segmentation with 3D CNNs) or tasks with different data modalities (e.g., tabular data, natural language processing).
Unverified key hypotheses: The paper assumes a direct link between "parameter deviation (global-local parameter difference) and the need for personalization"—a core premise of the OBP score. However, this link is treated as an observation rather than a validated mechanism. For example, the paper does not explain why parameters with larger deviations are more critical for personalization, or how this deviation correlates with local data distribution shifts. Formalizing and validating this hypothesis would strengthen the method’s theoretical foundation.

**Questions:**

- Regarding the OBP approximation error: Have you analyzed how multi-step local updates (e.g., 10 or 20 local epochs, instead of 5) or biased aggregation (e.g., FedProx with varying regularization strengths) affect the accuracy of the OBP score? If the approximation error increases in these scenarios, do you have strategies to correct it (e.g., incorporating cumulative gradients)?
- For the threshold q: Have you explored adaptive strategies to adjust q dynamically (e.g., based on local validation loss, client data heterogeneity α, or global model convergence)? If so, what performance trends did you observe? If not, do you believe an adaptive q would be feasible, and what metrics would you prioritize for its design?
- On experimental generalization: Do you have plans to test FedOBP on more complex architectures (e.g., ResNet-50, ViTs) or non-classification tasks, e.g., medical image segmentation)? For architectures with batch normalization (known to be problematic in non-IID FL), how would FedOBP handle personalized parameters in normalization layers?
- Regarding the second-order term: Have you attempted to compute the second-order term (Hessian diagonal entries) for a subset of parameters to quantify its magnitude relative to the first-order term? This would provide theoretical justification for its neglect, rather than relying solely on empirical observations.

---

### Official Review · Reviewer_AYPL · 2025-11-01

**Soundness:** 1
**Presentation:** 2
**Contribution:** 2
**Rating:** 2
**Confidence:** 3

**Summary:**

This paper proposed `FedOBP`, a personalized federated learning algorithm based on parameter decoupling, of which importance is measured using the principle inspired by the optimal brain damage pruning theory.

**Strengths:**

- While parameter decoupling-based PFL methods are saturated, optimal brain damage pruning theory-inspired idea itself is new and innovative.
- The empirical success in saving communication cost is impressive.

**Weaknesses:**

- While the adoption of OBP in FL is novel, the process of its adoption skips logical and theoretical evidence (from Eq. (14) to Eq. (19)).
  - The OBP is no more than local quadratic approximation of a global model w.r.t. local model for all dataset.
  - However, it is not convincing how this approximation can be approximated again to be a squared norm of L2 difference between the global and local models.
  - Moreover, evaluation of a local model on all dataset is itself not making sense.
  - As it is a main contribution of the proposed method, this logical flaw is critical and should have been addressed clearly.
- There is no theoretical guarantee on convergence or generalization guarantee.
  - It is required in that reported empirical results show questionable behavior.
  - In Figure 2 and 3, while the final performances are decent, the saturation speed (i.e., slope of accuracy) of the proposed method is slower than other methods, questioning "fastest convergence" in line 399.
  - Moreover, there exist "hops" in CIFAR100 and SVHN benchmarks in Figure 2. These should be explained as they are clearly distinctive behaviors compared to others.

**Questions:**

- Please unify citation style: (Collins et al., 2022) vs. FedSelectTamirisa et al. (2024)
- $\subset$ -> $\subseteq$ and $\gamma\in(0,1]\text{ s. t. }\vert \mathcal{C}^t \vert = \max(1, \gamma \cdot \vert \mathcal{C} \vert)$ in line 99.
- "empirical loss" -> "expected loss" in line 102.
- How can we interpret the model aggregation in Eq. (11) as "performing a single
update step for the local models on the global dataset D", to reach Eq. (17)? (lines 303-304)
  - Without theoretical justification or plausible derivation, it seems crude approximation and not reasonable.
- In Eq. (18), why is the step size $\eta$ ignored? It should be explicitly considered as it is an important factor to control convergence of typical FL algorithms.
- How is the local model $\boldsymbol{\theta}_i^{t-1}$ initialized when if the client is elected the very first time? Same copy as a global model?
- Please provide in-depth analysis on the comparison between selected parameters and filtered parameters of the classification layer. (e.g., their impact on final accuracy, their difference in resulting activation, ...)
  - In Figure 5, it shows that the classification layer alone is sufficient for personalization.
  - While this is a classical assumption in some previous parameter decoupling-based PFL methods, as mentioned in lines 466–468, the derived conclusion could imply that unselected parameters would negatively impact personalization performance.
  - This could be analyzed more using CKA, which is already mentioned in the paper.

---

### Note · Authors · 2025-11-13

**Comment:**

We sincerely thank the reviewers for their valuable feedback and insightful comments. After careful consideration, we have decided to withdraw the current submission.

We plan to revise our work to address the concerns raised, particularly regarding experimental scalability, and methodological clarity.  Given the scope of required revisions, we believe this work exceeds the current review cycle's limitations. We appreciate the reviewers' insightful comments and hope to submit a improved version in the future.

**Withdrawal Confirmation:**

I have read and agree with the venue's withdrawal policy on behalf of myself and my co-authors.